# Short communication: Feasibility of dengue vaccine to infect different human cell lines: An alternative potency test using HEK293T cells

**Renata Faria de Carvalho**[1,2], **Lucas de Siqueira Penna Quintaes**[1], **Thaís de Cássia de Souza Su**[1], **Leticia Mitiko Kobayashi**[1], **Ana Cristina Martins de Almeida Nogueira**[2,3]*

**1** Viral Vaccines Laboratory, National Institute of Quality Control in Health, FIOCRUZ, Rio de Janeiro, Brazil, **2** Post-Graduation Program in Sanitary Surveillance, National Institute of Quality Control in Health, FIOCRUZ, Rio de Janeiro, Brazil, **3** Clinical Immunology Laboratory, Oswaldo Cruz Institute, FIOCRUZ, Rio de Janeiro, Brazil

* ana.martins@ioc.fiocruz.br

## Abstract

Dengue is caused by an arbovirus that belongs to the Flaviviridae family and there are four distinct, but close related, circulating serotypes. Dengue disease is of great importance for global public health, with vaccination being its main prophylactic measure. However, there is a paucity of biological models for evaluating tetravalent dengue vaccines. The aim of this study was to evaluate the susceptibility of human cell lines HEK293T and THP-1 to a commercial dengue vaccine and test the feasibility of this approach in the development of a potency assay with human cell lines, as a methodological alternative to the golden standard potency assay with VERO cells. In this context, we used a batch of the commercial vaccine Dengvaxia® (CYD-TDV) for the infection tests. We evaluated the presence of the vaccine virus in THP-1 cells, differentiated into macrophages (dTHP-1), and in HEK293T by confocal microscopy, using 4G2 pan-flavivirus antibody. Vaccine infectivity and potency were determined by immunocolorimetric assay using monoclonal antibodies specific for each serotype. The results indicated that the human strain HEK293T was responsive to the tetravalent vaccine, as shown by the presence of virus particles in the cell cytoplasm in a pattern similar to the one observed with VERO cells. Moreover, it was possible to determine the infectivity and potency values of each vaccine virus serotype in the HEK293T, with serotype 4 prevailing over the others. Thus, the human cell line HEK293T provides a potential candidate to be used in assays to determine potency and identity of tetravalent dengue vaccines.

## Introduction

Dengue, an arthropod-borne illness, occurring in tropical and subtropical areas, is an important public health burden, with approximately 40% of the world population at risk of infection [1]. It is estimated that 390 million cases of dengue virus (DENV) infection occur annually, while 96 million are clinically manifest [2]. DENV circulates as four distinct serotypes (DENV-1 to 4) and the illness can evolve from a mild dengue fever to a severe form. The severe manifestation of dengue disease is often hasty and it is known as dengue hemorrhagic fever/dengue shock

**Data Availability Statement:** All relevant data are within the paper and its Supporting Information files.

**Funding:** This work was supported by INCQS and IOC, both FIOCRUZ. ACMAN and RFC received institutional grant. This work was part of the doctoral thesis of RFC at the Post-Graduation Program in Sanitary Surveillance. The funders had no role in the study.

**Competing interests:** The authors have declared that no competing interests exist.

syndrome (DHF/DSS). This complication is manifested by thrombocytopenia, hemorrhagic symptoms and increased vascular permeability, leading frequently to death [3]. Due to the major epidemiological burden that dengue illness characterizes, the development of an effective vaccine against the etiological agent of this disease is considered a worldwide priority [4].

A dengue vaccine should protect against all four serotypes of DENV simultaneously, provide long-term protection, be safe and well tolerated. Additionally, antibody-dependent enhancement phenomenon (ADE) must be strictly avoided in any vaccine proposed for the prevention of DENV infection [5]. Up to the present day, Sanofi Pasteur´s Dengvaxia® tetravalent dengue vaccine (CYD-TDV) is the only one registered worldwide [6]. However, there are several other candidate tetravalent vaccines in preclinical and clinical development, such as the TV003/TV005 (US NIH/Brazil Butantan Institute) and DENVax (Japan Takeda laboratory) [7].

Robust and reproducible production processes guarantee the quality of vaccines. The recommended quality control tests that ensure the effectiveness of the dengue tetravalent vaccine are: identity, potency and thermostability [8]. Potency, as defined by the International Council for Harmonisation (ICH), is the ability of a product to generate a specific biological activity that can be quantified [9]. The determination of CYD-TDV potency is based on a 50% cell culture infective dose assay ($CCID_{50}$) essentially performed in VERO cells (African green monkey kidney epithelial cells). However, as established by WHO, the potency of tetravalent vaccines against dengue can also be determined on other cell lineages, as soon as these cells are susceptible to the vaccine virus [10].

Assays performed in VERO cells demonstrated that the CYD-TDV resulted in a balanced viral and antibody titers for the four viral serotypes of DENV [11–15]. Immunogenicity studies, however, point to unbalanced viral and antibody titers, with a strong dominance of serotype 4 (DENV-4) over the other serotypes (DENV-1, DENV-2 and DENV-3) [16–19]. Despite the historical application of VERO cells to determine infectivity of virus, those findings underlined the necessity of considering other cell lineages that would still be passive of infection, but also able to point differences in response to each DENV serotype. Therefore, we aimed to investigate the infectivity, along with the determination of potency of CYD-TDV in two different human cell lines and then to compare the results obtained with the human lines with the VERO cell assay. For that purpose, we have chosen a monocyte/macrophage like human leukemic cell lineage, THP-1, due to the fact that those cells respond to wild DENV infection, as published elsewhere. Besides, as a macrophage like cell, it is closer to the physiological situation than VERO cells, since it may act as an antigen presenting cell [20]. The second cell line elected to this study was the human embryonic kidney 293T lineage, HEK293T. That cell is widely used in virus research, because of its high susceptibility to transfections, easy genetic manipulation and permissibility to virus infection [21]. Out of those facts, we stood by the motivating hypothesis that THP-1 (differentiated THP-1 or dTHP-1) and HEK293T cells lineages, other than VERO cells, can be used to determine differences in the infectivity of each serotype of CYD-TDV. The results showed that infection of HEK293T cells was possible with all four vaccine serotypes. Furthermore, after establishing a vaccine dose-response curve in these cells, we observed that the potency accessed in HEK293T cells varied for each serotype, while no significant variation was detected in VERO cells.

## Materials and methods

### Cell culture

VERO cells (ATCC Cat No. CCL-81) were cultured at 37˚ C ± 1˚ C and 5% $CO_2$ in MEM (Gibco) supplemented with 5% FBS (Hyclone, Thermo Scientific), 2 mM L-glutamine

(Gibco), 26 mM NaHCO$_3$ (Eurobio) and 100 U/mL of penicillin and 100 μg/mL of dihydros-treptomycin (Eurobio). For the infectivity assays, VERO cells were kept in the same culture medium but supplied with 10% FBS. 2% HEPES were added to the cell medium once cells were exposed to the vaccine dilutions.

The HEK293T cells were kindly donated by Prof Dr Raquel Ciuvalschi Maia from the Laboratory of Cellular and Molecular Hemato-Oncology (INCA, Rio de Janeiro, Brazil). They were cultured in DMEM (Sigma-Aldrich) supplemented with 10% FBS (Hyclone, Thermo Scientific), 2 mM L-glutamine (Gibco), 26mM NaHCO$_3$ (Eurobio), 25 mM HEPES (Gibco), 100 U/mL of penicillin and 100 μg/mL of dihydrostreptomycin (Eurobio), at 37°C ± 1°C and 5% CO$_2$.

THP-1 cells, obtained at Rio de Janeiro Cell Bank—BCRJ—accession number 0234, were cultured in RPMI-1640 medium (Sigma-Aldrich) supplemented with 10% FBS (Hyclone, Thermo Scientific), 26 mM NaHCO$_3$ (Eurobio), 100 U/mL of penicillin and 100 μg/mL of dihydrostreptomycin (Eurobio), 1 mM of sodium pyruvate (Gibco), at 37°C ± 1°C and 5% CO$_2$. For the infectivity assays, THP-1 cells were previously treated with 20 ng/mL of PMA for 48 hours to obtain macrophage-like THP-1 cells (dTHP-1), as described elsewhere [22].

## Vaccine and antibodies

Commercially available dengue vaccine (Dengvaxia® from Sanofi Pasteur, France) was used in all experiments described in this manuscript. The pan-flavivirus antibody (4G2), kindly provided by BioManguinhos (FIOCRUZ, Rio de Janeiro, Brazil), was used for determining the positivity of infection at the immunofluorescence assay. Determination of vaccine dose-response and potency were performed using individually murine anti-Envelope DENV protein monoclonal antibodies (mAb) from each serotype (anti-CYD1 D2-1F1-3, Batch No. VT150630-3176; anti-CYD2 3H5-1-12 Batch No. VT170329-4270; anti-CYD3 8A1-2F12, Batch No. VT150713-3176; anto-CYD4 1H10-6-7, Batch No. VT150701-3176), kindly donated by Sanofi Pasteur, and the polyclonal goat anti-mouse IgG secondary antibody conjugated to human alkaline-phosphatase (Lot. No. E2518-PK49F; Cat. No. 1030–04; SouthernBiotech).

## Immunofluorescence

HEK293T, dTHP-1 and VERO cells were seeded to the Nunc® Lab-Tek® (Thermo Scientific) cell culture 8 chamber slides in concentrations of 3x10$^5$ cells/mL, 1x10$^6$ cells/mL and 3x10$^5$ cells/ml, respectively. Then, 100 μL of dengue vaccine in 10$^{-1}$ dilution was added to the chambers and the slides were incubated for 7 days at 36°C ± 1°C, 5% ± 1% CO$_2$.

After the incubation period, culture medium was discarded and the cells were gently washed with 200 μL of PBS (Eurobio). Then, the cells were fixed with 100 μL of acetone (85% v/v; Carlo-Erba Reagents) for 2 minutes at room temperature. After fixation, the cells were gently washed three times with 100 μL of PBS and then incubated for 20 minutes at 5°C ± 3°C in 100 μL of PBS / BSA 1% (Sigma-Aldrich) for blocking. After that, cells were labeled with 60 μL of 4G2 antibody conjugated to Alexa-Fluor 647, at a 1:200 dilution in 1% PBS / BSA + 0.1% Triton X-100 solution (Sigma-Aldrich), for 1 hour at 5°C ± 3°C, under light protection. After labeling, the cells were gently washed twice with 100 μL of PBS and then the slide culture chambers were removed. Then, 10 μL of Hoechst's dye 33258 (Sigma-Aldrich), for nucleus staining, and 5 μL of oil were added to each condition.

The slides were analyzed using a confocal microscope (Leica TCS-SP8) in 200x optical magnification. The analyzes were performed using the LAS X software.

## Immunocolorymetry

To assess the infectivity of dengue vaccine, 100 μL of HEK293T and VERO cells, at a concentration of $3x10^5$ cells/mL, were seeded in a 96-well microplate (Falcon). Then, in 10 replicates, 100 μL of the vaccine dilutions, prepared with culture medium, was added at concentrations of $10^{-1}$, $10^{-2}$ and $10^{-2.6}$, and 100 μL of culture medium was added as negative control. The plates were incubated for 7 days at 36° C ± 1° C and 5% $CO_2$.

To determine dengue vaccine titer, 100 μL of HEK293T and VERO cells at a concentration of $3x10^5$ cells/mL were seeded in a 96-well microplate (Falcon). Then, in 10 replicates, 100 μL of the vaccine was added to the wells in dilutions of $10^{-1.3}$ to $10^{-3.4}$, 1:2 dilution factor, for HEK293T cells, and $10^{-2.6}$ to $10^{-6.8}$, 1:4 dilution factor, for VERO cells. 100 μL of culture medium was added to negative controls. The plates were incubated for 7 days at 36° C ± 1° C and 5% $CO_2$.

After incubation, the cells were fixed with 250 μL of acetone (85% v/v) for 15 minutes at -20° C. After fixation, the plates were left to dry and, after dryness, 100 μL of saturation buffer (PBS with 2.5% skimmilk (Difco™ BD), 0.5% Triton X-100 (Sigma-Aldrich), 0.05% Tween-20 (Merck)) was distributed to the wells, followed by a 30 minutes incubation at 36° C ± 1° C. After that, saturation buffer was removed and 50 μL of primary anti-DENV-1, anti-DENV-2, anti-DENV-3, anti-DENV-4 diluted to a concentration of 1/5000 in saturation buffer, was added to the wells and plates were incubated for 45 minutes at 36° C ± 1° C. Then, the plates were washed twice with washing buffer (PBS with 0.05% Tween-20) and 50 μL of secondary antibody conjugated to alkaline-phosphatase, diluted to a concentration of 1/1000 in saturation buffer, was added to plates, followed by incubation for 45 minutes at 36° C ± 1° C. After incubation, the plates were washed twice and 50 μL of NBT/BCIP substrate (Sigma-Aldrich) was distributed to the wells, followed by incubation for 60–90 minutes at room temperature under light protection, while the pigmentation developed. After that, the substrate was removed, and the plates were washed twice with distilled water and left to dry.

The wells were evaluated for the presence of staining, using an inverted optical microscope (IMT, Olympus), at a magnification of 100X, indicating infection, for each dilution used. The $CCID_{50}$/dose values in $log_{10}$ were used to determine viral titers.

## Statistical analysis

Data is expressed as mean ± SD or SEM and evaluated with GraphPad prism software v5.1 (La Jolla, USA). To determine the linearity of the results found in the dose-response curve, a linear regression was performed for both dilutions used in VERO and HEK293T cells.

Viral titers were calculated by CombiStats® v5.0 program (European Directorate for the Quality of Medicines and HealthCare/Council of Europe, Strasbourg, France) using the Spearman & Karber method. Titers are expressed in $log_{10}$ $CCID_{50}$ per dose. To verify statistical differences in the potency of the different experimental groups, we applied the two-way ANOVA test with a post Bonferroni test. Statistical significant differences were defined with probability (p) values inferior or equal to 0.05.

## Results

### Infectivity of different human cell lines with DENV vaccine

To investigate whether a human cell line would be susceptible to infection by DENV vaccine, we chose two human derived cell lines, dTHP-1 and HEK293T. After 7 days of incubation with $10^{-1}$ DENV vaccine dilution, we performed immunofluorescence assay by confocal microscopy. VERO cells, widely used in infectivity tests, functioned as a positive control.

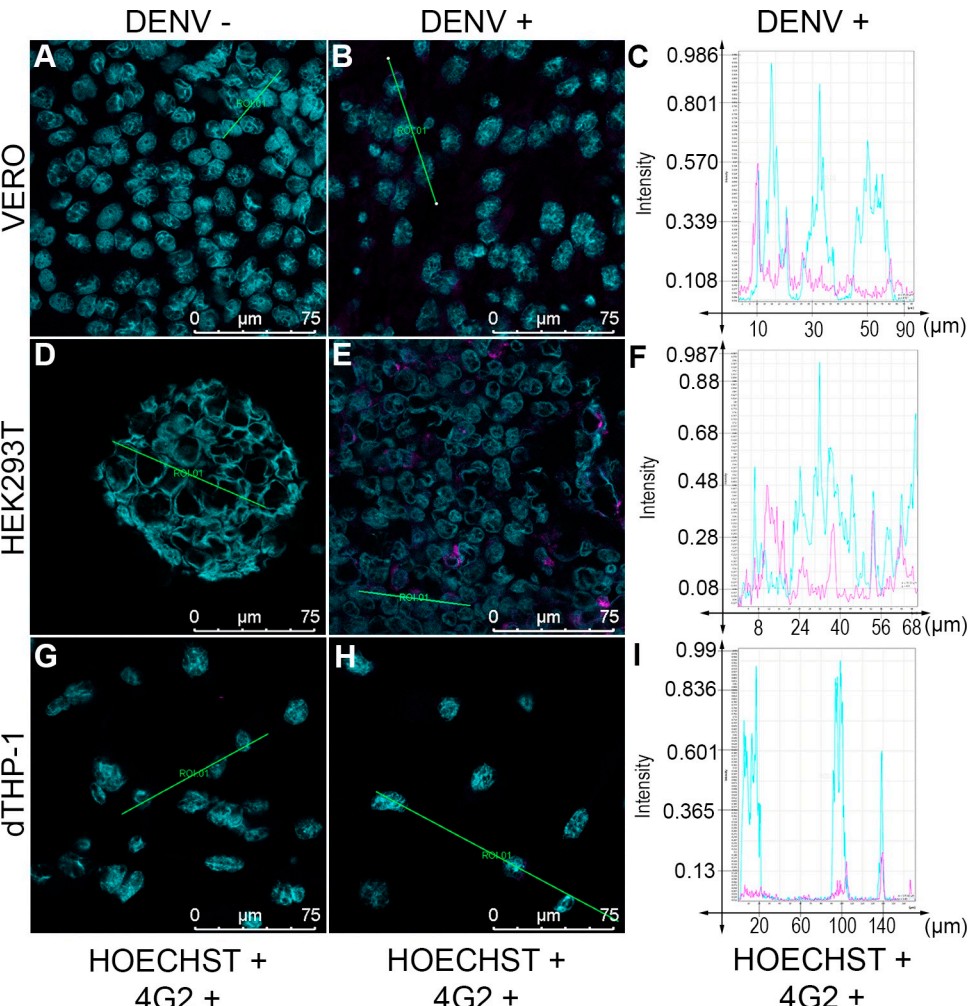

**Fig 1. Confocal immunofluorescence of VERO, HEK293T and PMA-treated THP-1 cells infected or not with dengue vaccine.** Cells were stained with anti-4G2 (pan-flavivirus antibody, pink fluorescence) and PAFI (DNA staining, blue fluorescence). Negative control (A), (D) and (G), infected cells with $10^{-1}$ vaccine dilution for 7 days (B), (E) and (H) show positivity for 4G2 in VERO (B) and HEK293T (E) cells. Representative graph of fluorescence intensity (y-axis) against the distance in micrometer (x-axis) of infected cells (C), (F) and (I). Representative images obtained under confocal microscope Leica TCS-SP8 at 200X magnification. n = 2.

As expected, we observed viral infectivity in the cytoplasm of VERO cells, as determined by the positive staining with pan-Flavivirus antibody 4G2 (Fig 1B). The fluorescence intensity increased considerably in infected VERO cells once compared with the controls (Figs 1C and S1). Regarding HEK293T cell line, a clear staining of 4G2 antibody detected in the cytoplasm of these cells, determined their susceptibility to infection with DENV vaccine, under our experimental conditions (Fig 1E). In contrast to the scattered fluorescence in the cytoplasm of VERO cells, the fluorescence signal detected in infected HEK293T cells was mostly evidenced in the perinuclear region (Figs 1F and S2). In the differentiated THP-1 cells, however, we did not detect any staining of 4G2 antibody as evidenced in both confocal image and graph analysis (Figs 1H, 1I and S3). As negative controls, cells were also stained with the 4G2 antibody in the absence of DENV vaccine and, as expected, showed no detectable signal (Fig 1A, 1D and 1G).

## Quantification of infection of DENV vaccine in VERO cells and in HEK293T cells

After determining the susceptibility of HEK293T to DENV vaccine, we tested whether these cells would respond to the same infectivity assay used in VERO cells to quantify infection and determine potency of DENV vaccine. For this purpose, we performed the infection assay with a $10^{-1}$ vaccine dilution and determined infectivity using a monoclonal antibody specific for dengue virus serotype 2 (CYD-2). VERO cell, positive control, and HEK293T infection assay was run together and precisely under the same conditions.

Both cells showed positivity to CYD-2 antibody, indicating that infection of HEK293T cells with at least vaccine virus serotype 2 was measurable using immunocolorimetric assay. The distinctive dark staining of infected cells (Fig 2B and 2D), when compared to non-infected cell controls, evidenced this finding (Fig 2A and 2C).

The next step was to obtain an optimal vaccine dilution curve, to confirm the feasibility of HEK293T cells to respond to different DENV vaccine dilutions. HEK293T cells were exposed

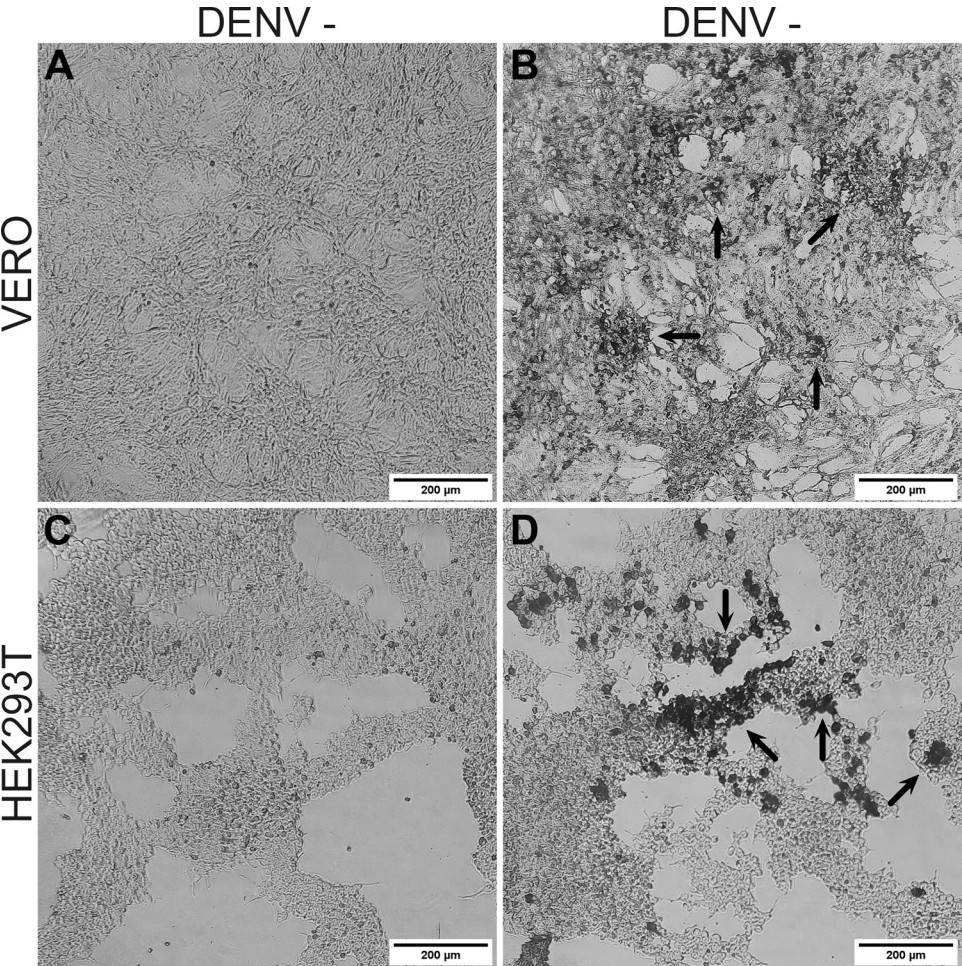

**Fig 2. Dengue vaccine infectivity in VERO and HEK293T cells by immunocolorimetric assay.** VERO and HEK293T cells incubated with dengue vaccine, $10^{-1}$ vaccine dilution, for 7 days. Control VERO cells (A) and infected VERO cells (B) stained with anti-E DENV-2 mAb. Control HEK293T cells (C) and infected HEK293T cells (D) stained with anti-E DENV-2 mAb. Black arrows indicate viral infection. Representative images obtained under Olympus microscope at 100X magnification. n = 4.

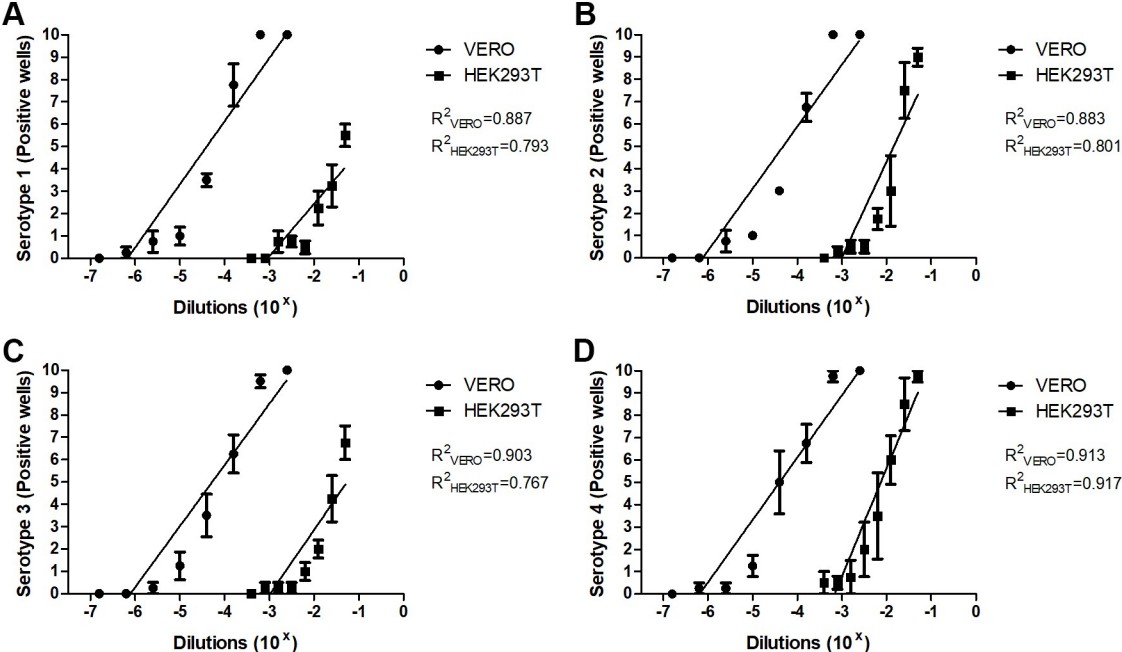

**Fig 3. Dose-response curve showing VERO and HEK293T cells response after 7 days of infection with Dengvaxia.** VERO (black circles) and HEK293T (black squares) cells infected with dengue vaccine (4,5–6,0 $\log_{10}$ $CCID_{50}$/dose) in dilutions of $10^{-2.6}$–$10^{-6.8}$, 1:4 dilution factor, and $10^{-1.3}$–$10^{-3.4}$, 1:2 dilution factor, respectively, and analyzed by immunocolorimetric assay after 7 days of incubation. Infection of serotype 1 (A), serotype 2 (B), serotype 3 (C) and serotype 4 (D) evaluated by the presence of staining after incubation with serotype-specific mAb—positive wells. Linear regression (straight lines) of infectious responses in VERO and HEK293T cells for serotypes 1 ($R^2_{VERO}$ = 0.887 and $R^2_{HEK293T}$ = 0.793), serotype 2 ($R^2_{VERO}$ = 0.883 and $R^2_{HEK293T}$ = 0.801), serotype 3 ($R^2_{VERO}$ = 0.903 and $R^2_{HEK293T}$ = 0.767) and serotype 4 ($R^2_{VERO}$ = 0.913 and $R^2_{HEK293T}$ = 0.917). Data presented as mean ± SEM. n = 4.

to different DENV vaccine dilutions from $10^{-1.3}$ to $10^{-3.4}$, following a dilution factor of 1:2, whereas VERO cells were subjected to dilutions from $10^{-2.6}$ to $10^{-6.8}$, with a dilution factor of 1:4, during 7 days of incubation. In this set of experiments, specific monoclonal antibodies for each serotype determined separately the degree of infectivity of the DENV vaccine both in VERO and HEK293T cells. We observed infection of HEK293T cells only at the most concentrated dilutions of the vaccine, as determined by the presence of dark color after incubation with monoclonal antibody specific to each dengue serotype (positive wells). Even though, the DENV vaccine dilutions used to infect VERO cells were less concentrated when compared to the ones used to infect HEK293T cells, VERO cells confirmed their higher susceptibility to the infection with this vaccine under our experimental conditions.

The dose-response curves and the linear coefficient of the regression equation of infectious responses in VERO and HEK293T cells for each serotype are shown in Fig 3 (serotype 1 $R^2_{VERO}$ = 0.887 and $R^2_{HEK293T}$ = 0.793, Fig 3A; serotype 2 $R^2_{VERO}$ = 0.883 and $R^2_{HEK293T}$ = 0.801, Fig 3B; serotype 3 $R^2_{VERO}$ = 0.903 and $R^2_{HEK293T}$ = 0.767, Fig 3C and serotype 4 $R^2_{VERO}$ = 0.913 and $R^2_{HEK293T}$ = 0.917, Fig 3D). All regression results showed over 75 percent of probability, thus supporting the idea that a dose curve response for the DENV vaccine in HEK293T cells is feasible.

## Potency of the 4 different DENV vaccine serotypes evaluated in HEK293T and VERO cells

One important feature regarding the assessment of the quality of a vaccine, whether in the developmental or in the control phase, is to verify the potency. Therefore, the final step of this

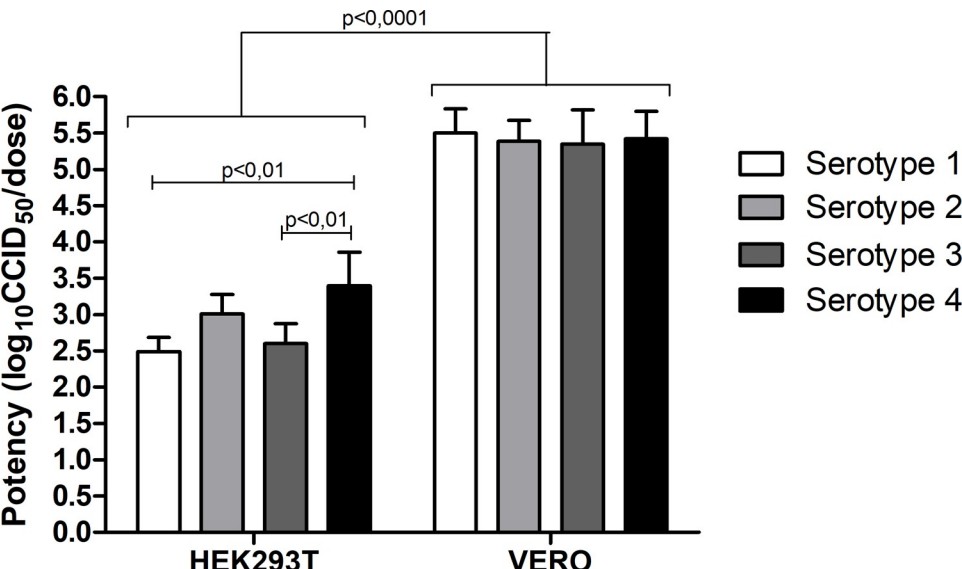

**Fig 4. Comparison of the infectivity of the four distinct DENV serotypes in VERO and HEK293T cells.** $CCID_{50}$ (mean ± SD) of each serotype measured by immunocolorimetric assay after 7 days of infection. Two-Way ANOVA performed for group comparison and significant differences observed among cell types ($p<0.0001$) and serotypes 4 and 3 ($p<0.01$) and serotypes 4 and 1 ($p<0.01$) measured in HEK293T cells. No significance detected in VERO cell groups. n = 4.

experimental set up was to calculate the estimated potencies among the 4 different serotypes in the same cell line and compare the results obtained in VERO cells with the ones in HEK293T cells. The result of the calculated potency, illustrated in Fig 4 in $log_{10}$ $CCID_{50}$/dose, in HEK293T cells for each serotype was 2.49 ± 0.19 (CYD-1), 3.01 ± 0.26 (CYD-2), 2.6 ± 0.27 (CYD-3) and 3.39 ± 0.46 (CYD-4) and in VERO cells, the estimated potencies were 5.49 ± 0.33 (CYD-1), 5.38 ± 0.28 (CYD-2), 5.34 ± 0.47 (CYD-3) and 5.42 ± 0.37 (CYD-4). Interestingly, we observed a different behavior on the potency estimated for each serotype using VERO or HEK293T cells. Whereas in VERO cells the estimated potency was approximately the same for each serotype, in HEK293T cells the serotypes showed different results. Regarding serotype 4, the potency showed a significantly higher result when compared to the ones detected for serotypes 1 and 3 in HEK293T cells ($p<0.01$). Additionally, the serotype 2 showed a slightly higher potency result when compared to the one obtained with serotype 1 and 3, but this tendency wasn´t statistically significant (Fig 4).

Comparisons of the estimated potencies for each serotype in VERO and HEK293T cell lines demonstrated that DENV vaccine achieved a significant higher potency result in the usual VERO cell model once compared to the HEK293T cell model proposed here ($p<0.0001$; Fig 4).

## Discussion

In this work, we investigated the feasibility of dengue vaccine to infect two human cell lines and subsequently the viability of these cells in estimating each serotype potency as an alternative or a complement to the VERO cells assay.

VERO cells are widely used in the development and control of vaccines due to their low cost and easy handling, but above all features one should stress that they are highly susceptible to infection by most viruses [23]. As matter of fact, once the addressed problem regards virus titers, to sought for a highly permissive cell is certainly a necessity. However, since this high

permissibility may possibly be associated to the absence of an effective antiviral response, this ought to be considered before drawing conclusions from data obtained solely with those cells. In fact, VERO cells lack antiviral response mediated by type 1 interferons [23,24]. If this response is by some means important for the experimental approach, the use of VERO cells alone might lead to misinterpretation of the results.

In the context of dengue, VERO cells are used in the production of attenuated virus tetravalent vaccines due to the high susceptibility to DENV [25,26]. Despite its widespread use, the high susceptibility to a heterologous virus vaccine may lead to overestimated infectivity and therefore give as an outcome not necessarily tangible potency value of each serotype, which may turn out not to be consistent to observations *in vivo* [15–17]. To test our hypothesis, we evaluated the responsiveness of HEK293T and THP-1 cell lines to the attenuated viral vaccines by measuring their susceptibility to infection and analyzing virus vaccine serotypes individually.

HEK293T cells have been used in the research, production and control of viral vaccines, mainly viral vector and nucleic acid vaccines, for example, the COVID-19 and Influenza vaccines, respectively [27–29]. Regarding dengue, HEK293T cells were shown to be susceptible to infection by wild DENV and other members of the Flaviviridae family [30]. Both the report that HEK293T is used in control trials of virus vaccines and that it is susceptible to wild dengue virus, led us to consider these cells as a candidate for our proposal. In fact, in the present study, we showed that the vaccine viruses constituting the tetravalent vaccine against dengue (Dengvaxia®) yield infection in these cells, under our experimental conditions, as proven by the observation of viral particles in cell cytoplasm. To our knowledge, this is the first report showing the susceptibility of these cells to DENV vaccine. In many aspects, this is an interesting finding, since it also indicates potential novel paths for studying mechanisms of action of Dengvaxia®. But for our purpose in this study, we needed to comparatively evaluate the use of HEK293T and VERO cells to measure potency. The determination of vaccine potency consists of evaluating the response of the vaccine product to a given biological substrate in order to estimate the potential of a vaccine to generate the desired final effect [31]. For attenuated vaccines, potency is generally expressed in terms of infectious units contained in a dose, as established in clinical studies [32]. According to WHO, for tetravalent vaccines against dengue, potency must be evaluated in terms of individual titers of each of the four serotypes in viral titration assays by plaque, $CCID_{50}$ or immunofocus in culture of VERO cells or other sensitive cells [32]. Since VERO cells are the golden standard in this field, the sought of another cell line might be performed in comparison with them.

Our results demonstrated that the HEK293T cell line is permissive to infection by the Dengvaxia® vaccine, even though, as expected, less susceptible than the VERO cell line. Hence, this study showed the viability of HEK293T cells as an alternative or complementary cell line for determining infection of DENV vaccine in different concentrations, using the immunoassay recommended by the vaccine producer. As a matter of fact, the dose-dependent curve obtained with these cells was found to be similar to the one observed with VERO cells and, therefore, this model could be recommended as an additional cell model.

Moreover, by using HEK293T cells it was possible to determine potency values for each of the serotypes present in the vaccine, in $CCID_{50}$ values, as recommended by WHO. Interestingly enough, HEK293T cells responded differently for each virus vaccine serotype with respect to their potency levels, as higher values were obtained associated to serotype 4 when compared to the others. VERO cells, in contrast, showed no significant differences between the virus vaccine serotypes regarding potency levels. In both pre- and clinical trials concerning the immunogenicity to the vaccine, a dominance of antibodies against the serotype 4 in comparison to the others was observed in the serum samples of vaccinated non-human primates and volunteers [16,17], despite the tests of potency in VERO cells had shown no differences

between the serotypes. Some studies, investigating the virus vaccine viremia of Dengvaxia® in individuals serum negative, showed a predominance for the CYD-4 and highest titer of DENV-4 neutralizing antibodies after vaccination. Regarding the vaccine TV003/TV005 from US NIH/Butantan Institute, consisted of a chimera of DENV-2 attenuated virus (backbone DENV-4) and attenuated DENV-1, DENV-3 and DENV-4, different formulations were tested in an attempt to balance the serotypes antibody titers. Thus, this study demonstrated that balanced infectivity correlated with the production of homotypic antibody [33]. Another vaccine approach developed by Takeda, used DENV-2 as the backbone virus to build a chimeric tetravalent vaccine. Again, differences in seroconversion were detected, but in this case, DENV-2 immunogenicity was higher once compared to the other serotypes. Alteration of concentrations of the vaccine virus like particles lead to a shift on the response. However, the increase of one specific serotype concentration–DENV-4 –with respect to the others resulted in a decrease on the seroconversion of DENV-1 serotype [34]. Yet, though the concentrations play an important role on seroconversion, other factors like cellular response and the virus strategy in the vaccine formulations have also to be explored [35].

Indeed, it might be helpful to use a cell that may better predict infectivity fluctuations and, consequently participate to the development of more uniform vaccines regarding their immunogenicity. Even though our work is a first effort on this token, it already shows that HEK293T might be interesting for testing heterologous DENV vaccines, for their ability to measure differences in serotypes infectivity. Whether this cell could also be a tool for studying cellular responses remains an open subject.

Our hypothesis was also explored by testing another cell line, the THP-1 cells. Since these cells can be differentiated to macrophages and macrophages are target cells on a natural DENV infection [36], we wanted to verify whether these cells would be permissive to vaccine virus as well. In fact, the THP-1 cell line has been widely used to study the mechanisms of DENV infection [37–39]. We observed no cellular susceptibility of differentiated THP-1 to infection by the CYD-TDV tetravalent vaccine at the time and vaccine concentrations studied. The absence of infection might be related to viral attenuation, which in turn leads to a loss in the virulence of vaccine viruses, added to the resistance of these cells to infections by DENV [34–41]. Herein, we concluded that THP-1 cells might be very useful for mechanism elucidation purposes rather than for potency assays.

The importance of developing vaccines against dengue capable of generating a balanced and effective immune response against the four viral serotypes, indicates the necessity of applying more sensitive *in vitro* and *in vivo* models during the establishment of those vaccines. In our work, we demonstrated the feasibility of CYD-TDV to infect the HEK293T cell line and that potency could be determined. Therefore, we considered the HEK293T cell line as a promising candidate in the development of potency assay and identity of tetravalent vaccines against dengue. As far as the simultaneous observation of different potencies for each serotype might be very useful in the development and production of heterologous dengue vaccine, our results reinforce the eligibility of HEK293T cell line as the basis for forthcoming assays.

## Supporting information

**S1 Fig. Confocal immunofluorescence graph of VERO cells infected with dengue vaccine.**
Cells were stained with anti-4G2 (pan-flavivirus antibody, pink fluorescence) and PAFI (DNA staining, blue fluorescence). Representative graph of fluorescence intensity (y-axis) against the distance in micrometer (x-axis) of infected VERO cells, obtained from confocal representative image (confocal microscope Leica TCS-SP8 at 200X magnification). n = 2.
(TIF)

**S2 Fig. Confocal immunofluorescence graph of HEK293T cells infected with dengue vaccine.** Cells were stained with anti-4G2 (pan-flavivirus antibody, pink fluorescence) and PAFI (DNA staining, blue fluorescence). Representative graph of fluorescence intensity (y-axis) against the distance in micrometer (x-axis) of infected HEK293T cells, obtained from confocal representative image (confocal microscope Leica TCS-SP8 at 200X magnification). n = 2.
(TIF)

**S3 Fig. Confocal immunofluorescence graph of PMA-treated THP-1 cells infected with dengue vaccine.** Cells were stained with anti-4G2 (pan-flavivirus antibody, pink fluorescence) and PAFI (DNA staining, blue fluorescence). Representative graph of fluorescence intensity (y-axis) against the distance in micrometer (x-axis) of infected PMA-treated THP-1 cells, obtained from confocal representative image (confocal microscope Leica TCS-SP8 at 200X magnification). n = 2.
(TIF)

**S1 Data.**
(XLSX)

## Acknowledgments

We are grateful to Dr Arnon Jurberg, from Universidade Estácio de Sá (UNESA), for his assistance with the immunofluorescence analyses. We also thank Dr Raquel Ciuvalschi Maia, from Instituto Nacional de Câncer (INCA), and Diagnostic Technology Laboratory (Lated/Biomanguinhos, Oswaldo Cruz Foundation), for the donation of HEK293T cell line and antibodies, respectively. In frame of our strong collaboration, we thank Sanofi Pasteur for kindly providing specific monoclonal antibodies against dengue serotypes.

## Author Contributions

**Conceptualization:** Renata Faria de Carvalho, Ana Cristina Martins de Almeida Nogueira.

**Data curation:** Renata Faria de Carvalho, Lucas de Siqueira Penna Quintaes, Thaís de Cássia de Souza Su, Leticia Mitiko Kobayashi, Ana Cristina Martins de Almeida Nogueira.

**Formal analysis:** Renata Faria de Carvalho, Lucas de Siqueira Penna Quintaes, Thaís de Cássia de Souza Su, Leticia Mitiko Kobayashi.

**Funding acquisition:** Ana Cristina Martins de Almeida Nogueira.

**Investigation:** Renata Faria de Carvalho, Ana Cristina Martins de Almeida Nogueira.

**Methodology:** Renata Faria de Carvalho, Lucas de Siqueira Penna Quintaes, Thaís de Cássia de Souza Su, Leticia Mitiko Kobayashi.

**Project administration:** Ana Cristina Martins de Almeida Nogueira.

**Resources:** Ana Cristina Martins de Almeida Nogueira.

**Supervision:** Ana Cristina Martins de Almeida Nogueira.

**Writing – original draft:** Renata Faria de Carvalho, Lucas de Siqueira Penna Quintaes, Thaís de Cássia de Souza Su, Leticia Mitiko Kobayashi, Ana Cristina Martins de Almeida Nogueira.

**Writing – review & editing:** Renata Faria de Carvalho, Ana Cristina Martins de Almeida Nogueira.

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
