## [Decision Letter · Decision Letter 0]

11 Feb 2022

PONE-D-21-33336Short Communication: Feasibility of dengue vaccine to infect different human cell lines: an alternative potency test using HEK293T cellsPLOS ONE

Dear Dr. Martins de  Almeida Nogueira,

Thank you for submitting your manuscript to PLOS ONE. After careful consideration, we feel that it has merit but does not fully meet PLOS ONE’s publication criteria as it currently stands. Therefore, we invite you to submit a revised version of the manuscript that addresses the points raised during the review process.

Comments of the Academic Editor:The rationale of this study needs to be clarified and the translational applications of this manuscript discussed in detail. 

Further points:

Stock solution for dilution in Figure 3 needs to be defined in the legend for Figure 3.

Figure needs to be high resolution and  have readable X- and Y- axis units (Figure 1)

Figure legends need to report number of n.

More details are needed regarding statistical analysis.

English writing style may be improved.

We look forward to receiving your revised manuscript.

Kind regards,

Henning Ulrich

Academic Editor

PLOS ONE

Journal Requirements:

“This work is part of doctor thesis in Post-Graduation Program in Sanitary Surveillance, INCQS (Oswaldo Cruz Foundation, Rio de Janeiro, Brazil).”

 “This work was supported by INCQS and IOC, both FIOCRUZ. ACMAN and RFC received institutional grant. This work was part of the doctoral thesis of RFC at the Post-Graduation Program in Sanitary Surveillance. The funders had no role in the study.”

Reviewers' comments:

Reviewer's Responses to Questions

**Comments to the Author**

1. Is the manuscript technically sound, and do the data support the conclusions?

Reviewer #1: Yes

Reviewer #2: Yes

2. Has the statistical analysis been performed appropriately and rigorously? 

Reviewer #1: N/A

Reviewer #2: I Don't Know

3. Have the authors made all data underlying the findings in their manuscript fully available?

Reviewer #1: Yes

Reviewer #2: Yes

4. Is the manuscript presented in an intelligible fashion and written in standard English?

Reviewer #1: Yes

Reviewer #2: Yes

5. Review Comments to the Author

Reviewer #1: The authors have submitted a manuscript describing the investigation of alternative human cell lines for dengue fever vaccine testing. This is a pure cell culture experiment without fresh human cells. Unfortunately, characteristics of the cell lines used are presented only incompletely and scattered throughout the paper, making the rationale behind the work very slow to emerge.

The introduction clearly shows that there is a need for tools to study new vaccines. Appropriate literature is used to explain the approach. The importance of the antibody response is hardly presented, here prior knowledge of the reader is required.

Materials and methods and results are well described. In the discussion, the authors conclude that the HEK293T cell is suitable as a model.

Reviewer #2: The present study provided interesting data of the infectivity and potency of the 4 serotypes used to formulate the DengVaxia vaccine in HEK293T cells when compared to Vero cells, though the group also shows the lack of infectivity of these serotypes on THP-1 cells differentiated to macrophages by PMA.

In fact, the more relevant results is related to the potency studies using HEK293T cells when compared to Vero cells that lacks the Type-1 interferon associated responses. As expected, the potency of the 4 serotypes were similar and higher in Vero cells when compared to HEK293T cells. Interestingly, the group could detect differences in HEK293T cells potency within the four serotypes.

However, the group should discuss in deep how this finding could be translated in a more adequate formulation that would be more close to what could be the in the clinical use of the vaccine, a equilibrated and balanced response against all the four serotypes. Based on the potency results, is it possible to predict a better formulation that would result in a more balanced response against the four serotypes? What would be the related amount of each serotype in vaccine formulation based on the HEK293T potency studies? Of course this can be validated by non-human primate immunization studies showing or not a equilibrated response against all the four serotypes in the new formulation based on the the potency studies on HEK293T cells in contrast to a more pronounced antibody production against the serotype 4 as found in the current vaccine formulation that was based on the potency studies based on Vero cells that presents a clear bias, as the result of the lack of Type-1 Interferon responses in these cells.

6. PLOS authors have the option to publish the peer review history of their article (what does this mean?). If published, this will include your full peer review and any attached files.

Reviewer #1: No

Reviewer #2: No

---

## [Author Response · Author response to Decision Letter 0]

21 Mar 2022

Response to the reviewers

PONE- D -21-33336

Comments of the Academic Editor and Reviewers

All the changes are highlighted in yellow in the MS file named “Revised manuscript with Track changes”. The pages and lines referred in the answers below are in that file to find.

1) The rationale of this study needs to be clarified and the translational applications of this manuscript discussed in detail. Answer: We added in the introduction part of the MS more to the rational of the work also regarding the cell lines used in the study (pg 4 and 5 Lines 89-99). In that section 2 new references are cited (refs number 20 and 21). In the discussion section, we considered the translational application of our findings, as suggested by both the Editor and the Reviewer 2. To this matter we added 3 new references (numbers 31, 32 and 33 ) and enhanced so the discussion (pg 18 -19 Lines 411-435).

2) Stock solution for dilution in Figure 3 needs to be defined in the legend for Figure 3. Answer: It is now added to the legend of Figure 3, pg 13.

3) Figure needs to be high resolution and have readable X- and Y- axis units (Figure 1) Answer: Figure 1 resolution is 300 dpi and the X and Y axis of Figure 1 C, F and I was added. That figure normal size is too big to fit in, so we added the scale manually and attached the original graph (Fig 1C ; Fig 1F; Fig 1I) as “Supporting Information Files” . 

4) Figure legends need to report number of n. Answer: All legends were revised, and n number was added when missing.

5) More details are needed regarding statistical analysis. Answer: We wrote the Statistical Analysis again (pg 9 lines 203-213).

6) English writing style may be improved. Answer: We revised the MS as to improve the English style. Some examples listed below: Pg 4 Line 80 and 81 was “point to an unbalanced in the viral and antibody titers produced” changed to “point to unbalanced viral and antibody titers”; Pg 16 Lines 361 was “ can super estimate infectivity and therefore give a not necessarily tangible potency value of each serotype” changed to “ may lead to overestimated infectivity and therefore give as an outcome not necessarily tangible potency value of each serotype”; Pg 18 line 397 was “ Thus” changed to “ As a matter of fact” same Pg line 400 was “Moreover, using HEK293T” changed to “ Moreover, by using …”: Pg 18 line 403 was “ vaccine serotype regarding…” changed to “ vaccine serotype with respect to …”; same page line 410 “ VERO cells showing no …” changed to “VERO cells had shown no…” 

7) In Acknowledgments we removed the statement “ This work is part of ….” as recommended, since it is already at the Funding Statement section. 

Reviewers' comments:

5. Review Comments to the Author

Reviewer #1: The authors have submitted a manuscript describing the investigation of alternative human cell lines for dengue fever vaccine testing. This is a pure cell culture experiment without fresh human cells. Unfortunately, characteristics of the cell lines used are presented only incompletely and scattered throughout the paper, making the rationale behind the work very slow to emerge.

The introduction clearly shows that there is a need for tools to study new vaccines. Appropriate literature is used to explain the approach. The importance of the antibody response is hardly presented, here prior knowledge of the reader is required.

Materials and methods and results are well described. In the discussion, the authors conclude that the HEK293T cell is suitable as a model.

Answer: To address the comment that the “characteristics of the cell lines used are presented only incompletely and scattered throughout the paper, making the rationale behind the work very slow to emerge”, we added in the Introduction a paragraph where we deeper characterized the cells (pg4 and 5 Lines 89-99) and we also included new references (refs number 20 and 21). 

Reviewer #2: The present study provided interesting data of the infectivity and potency of the 4 serotypes used to formulate the DengVaxia vaccine in HEK293T cells when compared to Vero cells, though the group also shows the lack of infectivity of these serotypes on THP-1 cells differentiated to macrophages by PMA.

In fact, the more relevant results is related to the potency studies using HEK293T cells when compared to Vero cells that lacks the Type-1 interferon associated responses. As expected, the potency of the 4 serotypes were similar and higher in Vero cells when compared to HEK293T cells. Interestingly, the group could detect differences in HEK293T cells potency within the four serotypes.

However, the group should discuss in deep how this finding could be translated in a more adequate formulation that would be more close to what could be the in the clinical use of the vaccine, a equilibrated and balanced response against all the four serotypes. Based on the potency results, is it possible to predict a better formulation that would result in a more balanced response against the four serotypes? What would be the related amount of each serotype in vaccine formulation based on the HEK293T potency studies? Of course this can be validated by non-human primate immunization studies showing or not a equilibrated response against all the four serotypes in the new formulation based on the the potency studies on HEK293T cells in contrast to a more pronounced antibody production against the serotype 4 as found in the current vaccine formulation that was based on the potency studies based on Vero cells that presents a clear bias, as the result of the lack of Type-1 Interferon responses in these cells.

Answer: In order to properly address the comment “However, the group should discuss in deep how this finding could be translated in a more adequate formulation….” , we added to the discussion 3 new references (Numbers 31, 32 and 33) on the theme and improved our discussion, as described in Pg 18 and 19 Lines 411-435. “Some studies, investigating the virus vaccine viremia of Dengvaxia® in individuals serum negative, showed a predominance for the CYD-4 and highest titer of DENV-4 neutralizing antibodies after vaccination. Regarding the vaccine TV003/TV005 from US NIH/Butantan Institute, consisted of a chimera of DENV-2 attenuated virus (backbone DENV-4) and attenuated DENV-1, DENV-3 and DENV-4, different formulations were tested in an attempt to balance the serotypes antibody titers. Thus, this study demonstrated that balanced infectivity correlated with the production of homotypic antibody [33]. Another vaccine approach developed by Takeda, used DENV-2 as the backbone virus to build a chimeric tetravalent vaccine. Again, differences in seroconversion were detected, but in this case, DENV-2 immunogenicity was higher once compared to the other serotypes. Alteration of concentrations of the vaccine virus like particles lead to a shift on the response. However, the increase of one specific serotype concentration – DENV-4 – with respect to the others resulted in a decrease on the seroconversion of DENV-1 serotype [34]. Yet, though the concentrations play an important role on seroconversion, other factors like cellular response and the virus strategy in the vaccine formulations have also to be explored [35]. 

Indeed, it might be helpful to use a cell that may better predict infectivity fluctuations and, consequently participate to the development of more uniform vaccines regarding their immunogenicity. Even though our work is a first effort on this token, it already shows that HEK293T might be interesting for testing heterologous DENV vaccines, for their ability to measure differences in serotypes infectivity. Whether this cell could also be a tool for studying cellular responses remains an open subject.”

---

## [Decision Letter · Decision Letter 1]

13 Apr 2022

Short Communication: Feasibility of dengue vaccine to infect different human cell lines: an alternative potency test using HEK293T cells

PONE-D-21-33336R1

Dear Dr. Martins de Almeida Nogueira,

We’re pleased to inform you that your manuscript has been judged scientifically suitable for publication and will be formally accepted for publication once it meets all outstanding technical requirements.

Kind regards,

Henning Ulrich

Academic Editor

PLOS ONE

Additional Editor Comments (optional):

Reviewers' comments:

Reviewer's Responses to Questions

**Comments to the Author**

1. If the authors have adequately addressed your comments raised in a previous round of review and you feel that this manuscript is now acceptable for publication, you may indicate that here to bypass the “Comments to the Author” section, enter your conflict of interest statement in the “Confidential to Editor” section, and submit your "Accept" recommendation.

Reviewer #1: All comments have been addressed

Reviewer #2: All comments have been addressed

2. Is the manuscript technically sound, and do the data support the conclusions?

Reviewer #1: Yes

Reviewer #2: Yes

3. Has the statistical analysis been performed appropriately and rigorously? 

Reviewer #1: Yes

Reviewer #2: Yes

4. Have the authors made all data underlying the findings in their manuscript fully available?

Reviewer #1: Yes

Reviewer #2: Yes

5. Is the manuscript presented in an intelligible fashion and written in standard English?

Reviewer #1: Yes

Reviewer #2: Yes

6. Review Comments to the Author

Reviewer #1: The authors show that HEK293T cell line a promising candidate in the development of potency assay and to identify vaccines against Dengue.

Reviewer #2: (No Response)

7. PLOS authors have the option to publish the peer review history of their article (what does this mean?). If published, this will include your full peer review and any attached files.

Reviewer #1: No

Reviewer #2: No

---

## [Editor Report · Acceptance letter]

29 Apr 2022

PONE-D-21-33336R1 

Short Communication: Feasibility of dengue vaccine to infect different human cell lines: an alternative potency test using HEK293T cells 

Dear Dr. Martins de Almeida Nogueira:

I'm pleased to inform you that your manuscript has been deemed suitable for publication in PLOS ONE. Congratulations! Your manuscript is now with our production department. 

Kind regards, 

on behalf of

Dr. Henning Ulrich 

Academic Editor

PLOS ONE